# Uniqueness of Minimax Strategy in View of Minimum Error Discrimination of Two Quantum States

**DOI:** 10.3390/e21070671

**Published:** 2019-07-09

**Authors:** Jihwan Kim, Donghoon Ha, Younghun Kwon

**Affiliations:** Department of Applied Physics, Hanyang University, Ansan, Kyunggi-Do 425-791, Korea

**Keywords:** quantum state discrimination, quantum minimax, uniqueness of strategy, guessing probability

## Abstract

This study considers the minimum error discrimination of two quantum states in terms of a two-party zero-sum game, whose optimal strategy is a minimax strategy. A minimax strategy is one in which a sender chooses a strategy for a receiver so that the receiver may obtain the minimum information about quantum states, but the receiver performs an optimal measurement to obtain guessing probability for the quantum ensemble prepared by the sender. Therefore, knowing whether the optimal strategy of the game is unique is essential. This is because there is no alternative if the optimal strategy is unique. This paper proposes the necessary and sufficient condition for an optimal strategy of the sender to be unique. Also, we investigate the quantum states that exhibit the minimum guessing probability when a sender’s minimax strategy is unique. Furthermore, we show that a sender’s minimax strategy and a receiver’s minimum error strategy cannot be unique if one can simultaneously diagonalize two quantum states, with the optimal measurement of the minimax strategy. This implies that a sender can confirm that the optimal strategy of only a single side (a sender or a receiver but not both of them) is unique by preparing specific quantum states.

## 1. Introduction

Quantum information processing can be achieved by discriminating quantum states, where classical information is encoded. Quantum states which are orthogonal to each other can be perfectly distinguishable. However, non-orthogonal quantum states cannot be perfectly discriminated. Therefore, one needs to have a discrimination strategy for non-orthogonal quantum states, and there are various strategies [1,2,3,4] such as minimum error discrimination (MD) [4,5,6,7], unambiguous discrimination [8,9,10,11,12], maximum confidence discrimination [13], and discrimination of fixed rate inconclusive result [14,15,16,17,18]. Unambiguous discrimination is a strategy where there is no error in the conclusive result by allowing an inconclusive result. Maximum confidence is a strategy where one maximizes the confidence of a conclusive result. Discrimination of fixed rate inconclusive result is a strategy where one may fix the rate of an inconclusive result. Among these strategies, the MD strategy can conclusively discriminate quantum states with a prior probability.

The MD strategy is employed for quantum states with a given prior probability, and the quantum states are optimally measured. MD strategy is that one maximizes the probability that the result of measurement of a receiver correctly points out the quantum state that a sender transmitted when only a conclusive result is permitted. The maximum probability is called guessing probability. One can investigate the behavior of MD in terms of a prior probability when quantum states are given.

Because the guessing probability is obtained based on prior probability, a change in prior probability results in different guessing probabilities, which implies that prior probabilities can be considered as a strategy of a sender. Even though one has discussed the uniqueness of measurement strategy in discrimination of two quantum states, the strategy of preparation such as a prior probability, which can be a strategy of a sender, has not been discussed in terms of identical guessing probability and optimal measurement strategy.

Quantum minimax approach is obtained by applying the minimax approach of a statistical decision to quantum state discrimination. Von Neumann, the inventor of game theory, showed that there exists a solution to the minimax problem when sender and receiver can choose a finite number of strategies in a two-person zero-sum game. Wald proved that the necessary and sufficient condition to the existence of a solution to the minimax problem is that the set of strategy for sender and receiver is countable [19]. Hirota and Ikehara discussed quantum minimax theorem, using the fact that the set of measurement strategy satisfies compactness [20]. They suggested the necessary and sufficient condition for minimax strategy in quantum state discrimination.

Further, by mean value theorem D’Ariano showed that there exists a quantum minimax strategy for two quantum state discrimination and provided a sufficient condition for the strategy [21]. However, in spite of these studies, the necessary and sufficient condition for uniqueness of minimax strategy in two quantum state discrimination is not known yet. Even more, the uniqueness of minimax strategy in two quantum state discrimination is not understood in terms of sender’s strategy, which is a selection of prior probability.

This study investigates a two-person zero-sum game where the payoff is defined by the correct probability of two quantum states [19,20,21,22]. The optimal strategy of the game is a minimax strategy, where the minimax strategy of a receiver is to select the optimal measurement providing MD and the minimax strategy of a sender is to choose the prior probability providing the minimum of guessing probability, which is displayed in Figure 1.

In this scenario, the prior probability and the measurement in MD are constructed as the strategy of a sender Alice and a receiver Bob [20,21]. First, Alice sends the quantum states, where classical information (x=1,2) is encoded, to Bob. Because the quantum states are not orthogonal to each other, a single measurement of Bob cannot perfectly discriminate the quantum states. Therefore, a suitable strategy is needed. Here Bob should choose a measurement strategy that can perform MD.

Meanwhile, a suitable selection of prior probability can be obtained by Alice, as a sender’s strategy. Alice’s strategy is to interfere with the minimum error strategy of Bob to minimize the guessing probability. Because Bob should perform MD without noticing Alice’s strategy, Bob tries to find an optimal strategy to obtain a payoff. Therefore, the minimum of guessing probability implies that a suitable selection of prior probability lets Bob obtain the minimum of guessing probability when Bob performs an optimal measurement. Furthermore, if Bob cannot perform an optimal measurement, he obtains a probability less than the guessing probability.

The quantum minimax theorem [20,21] can be used to prove that Alice and Bob can set up an optimal strategy on both sides. However, it is not known whether the minimax strategy is unique or not. The uniqueness of the optimal strategy of the game is important in performing the game. There is no alternative to a unique optimal strategy. Therefore, a strategy cannot be optimal if an error occurs when performing the strategy. However, a strategy can still be optimal if it is not unique, even though an error occurs in the strategy. In this light, it is crucial to know whether the minimax strategy is unique, when the strategy is optimal in the game. Here, we investigate the condition for uniqueness of the optimal strategy of a sender. The condition is described by the quantum states and the minimax strategy of a receiver. More explicitly, we study the necessary and sufficient condition for the uniqueness of a sender’s strategy. Using the condition, we investigate the quantum states that exhibit the minimum of guessing probability when a sender’s minimax strategy is unique.

Also, we show that a sender’s minimax strategy and a receiver’s minimum error strategy cannot be unique if two quantum states are simultaneously diagonalized with the optimal measurement of minimax strategy. Therefore, a sender can make the optimal strategy of only a single side unique by preparing specific quantum states. Our investigation can be applied to various fields. As the first example of our investigation, we explain how the BB84 protocol [23] with equal prior probability is optimal in terms of the minimax strategy. We also discuss how the results of this study can be applied to building a quantum random number generator(QRNG) [24,25,26].

This paper is organized as follows. In Section 2, we explain the necessary background of our investigation. In Section 3, for the minimax strategy of a sender, we provide the necessary and sufficient condition for uniqueness of the optimal strategy. We investigate the uniqueness of the strategy of the sender for some quantum states by using this condition. Furthermore, we obtain the condition under which both the sender’s minimax strategy and the receiver’s optimal minimum error strategy cannot be unique. Finally, we discuss the results and conclusions in Section 4.

## 2. Preliminaries

For two quantum states $\rho_{1}$ and $\rho_{2}$, the minimal subspace H for discriminating $\rho_{1}$ and $\rho_{2}$ should satisfy $H=\mathrm{Supp}(\rho_{1}+\rho_{2})$. In this study, we assume that the rank of quantum state is finite. Then, by the relation $\dim H\leq \mathrm{rank}\left(\rho_{1}\right)+\mathrm{rank}\left(\rho_{2}\right)$, a quantum state or an optimal measurement can be represented as an operator on finite dimensional Hilbert space.

The MD of two quantum states $\rho_{1}$ and $\rho_{2}$ is a strategy to determine the maximum value of correct probability $P_{\mathrm{corr}}=q\mathrm{tr}(\rho_{1}M_{1})+\left(1-q\right)\mathrm{tr}(\rho_{2}M_{2})$, which is called guessing probability, by performing an optimal measurement. The maximum value of the correct probability is known as Helstrom bound [27].

Assuming that the prior probabilities of two quantum states $\rho_{1}$ and $\rho_{2}$ are *q* and 1-q, respectively, one can obtain the following lemma in the MD of the two quantum states. (The proof can be found in the Appendix A).

**Lemma** **1**(Optimal condition of MD for two quantum states [27,28])**.**
*The necessary and sufficient condition for optimal measurement ${\left\{M_{x}\right\}}_{x=1}^{2}$ is given by*
(1)$${\left(-1\right)}^{x}\left(\left(1-q\right)\rho_{2}-q\rho_{1}\right)M_{x}\geq 0\;\forall x\in \left\{1,2\right\}.$$

In general, the optimal measurement in MD is not unique. If the nullity of operator $\Lambda \equiv \left(1-q\right)\rho_{2}-q\rho_{1}$ is *d*, there exist at least $2^{d}$ number of optimal extreme POVMs. A convex combination of these POVM also provides an optimal measurement of MD. When Λ has full rank, the optimal measurement is unique. Quantum minimax theorem tells that among optimal MD strategies there is at least a POVM of minimax strategy in a prior probability providing the minimum of guessing probability [20,21].

**Theorem** **1**(Quantum minimax theorem [20,21])**.**
*There exists an a priori probability $q^{★}=\left(q^{★},1-q^{★}\right)$ for the states $\rho_{1}$ and $\rho_{2}$, and a measurement $M^{★}=\left(M_{1}^{★},M_{2}^{★}\right)$ such that*
(2)$$\begin{array}{c} \min_{q}\max_{M}P_{\mathrm{corr}}\left(q,M\right)=P_{\mathrm{corr}}\left(q^{★},M^{★}\right)=\max_{M}\min_{q}P_{\mathrm{corr}}\left(q,M\right) \end{array}$$
*where $q^{★}\in \left(0,1\right),P_{\mathrm{corr}}\left(q,M\right)=\sum_{x=1}^{2}q_{x}\mathrm{tr}(\rho_{x}M_{x})$.*

Note that when quantum states are prepared in a prior probability q, $\max_{M}\min_{q}P_{\mathrm{corr}}\left(q,M\right)=P_{\mathrm{corr}}\left(q^{★},M^{★}\right)$ implies that the measurement of $M^{★}$ is optimal and $\min_{q}\max_{M}P_{\mathrm{corr}}\left(q,M\right)=P_{\mathrm{corr}}\left(q^{★},M^{★}\right)$ implies that the prior probability of $q^{★}$ provides the minimum of guessing probability. However, every optimal MD in the prior probability of $q^{★}$ is not a minimax strategy of Bob. Suppose that a measurement of $N=(N_{1},N_{2})$ in the prior probability of $q^{★}$ is an optimal strategy of MD, satisfying $\mathrm{tr}(\rho_{1}N_{1})>\mathrm{tr}(\rho_{2}N_{2})>0$. Then, the strategy of Alice in $q^{★}$ cannot be a prior probability for the minimax strategy, as $\tilde{q}=\left(0,1\right)$ of Alice’s strategy provides a lower guessing probability than that of $q^{★}$:(3)$$P_{\mathrm{corr}}\left(q^{★},N\right)=q^{★}\mathrm{tr}(\rho_{1}N_{1})+\left(1-q^{★}\right)\mathrm{tr}(\rho_{2}N_{2})>\mathrm{tr}(\rho_{2}N_{2})=P_{\mathrm{corr}}\left(\tilde{q},N\right).$$
Therefore, the first condition that the minimax strategy $M^{★}$ of Bob should satisfy is $\mathrm{tr}(\rho_{1}M_{1}^{★})=\mathrm{tr}(\rho_{2}M_{2}^{★})$. Because the measurement of $M^{★}$ is an optimal strategy for the prior probability of $q^{★}$, it satisfies the optimal condition of MD, which is the second condition. Inversely, the fulfillment of the two conditions is the sufficient condition for the minimax strategy.

Here, the conditions can be explained as follows. Suppose that a measurement $M^{\circ }=\left(M_{1}^{\circ },M_{2}^{\circ }\right)$ satisfies $\mathrm{tr}(\rho_{1}M_{1}^{\circ })=\mathrm{tr}(\rho_{2}M_{2}^{\circ })$ and is optimal for the prior probability of $q^{\circ }$. Then, we find the following relation:(4)$$\min_{q}\max_{M}P_{\mathrm{corr}}\left(q,M\right)\leq \max_{M}P_{\mathrm{corr}}\left(q^{\circ },M\right)=P_{\mathrm{corr}}\left(q^{\circ },M^{\circ }\right)=\min_{q}P_{\mathrm{corr}}\left(q,M^{\circ }\right).$$
The last equality holds by $\mathrm{tr}(\rho_{1}M_{1}^{\circ })=\mathrm{tr}(\rho_{2}M_{2}^{\circ })$. Because of $\min_{q}\max_{M}P_{\mathrm{corr}}\left(q,M\right)\geq \min_{q}P_{\mathrm{corr}}\left(q,M^{\circ }\right)$, we find $\min_{q}\max_{M}P_{\mathrm{corr}}\left(q,M\right)=\min_{q}P_{\mathrm{corr}}\left(q^{\circ },M^{\circ }\right)$. And the following relation holds:(5)$$\max_{M}\min_{q}P_{\mathrm{corr}}\left(q,M\right)\leq \min_{q}P_{\mathrm{corr}}\left(q,M^{\circ }\right)=P_{\mathrm{corr}}\left(q^{\circ },M^{\circ }\right)=\max_{M}P_{\mathrm{corr}}\left(q^{\circ },M\right)$$
The first equality is obtained by $\mathrm{tr}(\rho_{1}M_{1}^{\circ })=\mathrm{tr}(\rho_{2}M_{2}^{\circ })$. Because of $\max_{M}\min_{q}P_{\mathrm{corr}}\left(q,M\right)\geq \max_{M}P_{\mathrm{corr}}\left(q^{\circ },M\right)$, we obtain $\max_{M}\min_{q}P_{\mathrm{corr}}\left(q,M\right)=\max_{M}P_{\mathrm{corr}}\left(q^{\circ },M^{\circ }\right)$ and $\min_{q}\max_{M}P_{\mathrm{corr}}\left(q,M\right)=P_{\mathrm{corr}}\left(q^{\circ },M^{\circ }\right)=\max_{M}\min_{q}P_{\mathrm{corr}}\left(q,M\right)$. It implies that $\left(q^{\circ },M^{\circ }\right)$ is a minimax strategy. Then, one can obtain the following lemma.

**Lemma** **2.**
*When MD is performed for a given prior probability, the minimum of guessing probability is obtained iff an optimal measurement ${\left\{M_{x}\right\}}_{x=1}^{2}$ satisfies $\mathrm{tr}(\rho_{1}M_{1})=\mathrm{tr}(\rho_{2}M_{2})$.*


## 3. Results

This section presents the necessary and sufficient condition that ensures the uniqueness of the minimax strategy of a sender. Because there always exists a minimax strategy for the quantum minimax theorem, when one finds a minimax strategy, one can obtain the condition by which the strategy is unique. When MDs with different prior probabilities can provide the same guessing probability, the following lemma provides the condition by which the MDs with different prior probabilities can have the same optimal measurement (The proof of this lemma can be found in the Appendix A).

**Lemma** **3.**
*The quantum ensembles of $S_{1}$ and $S_{2}$ are given as ${\left\{p_{x},\rho_{x}\right\}}_{x=1}^{2}$ and ${\left\{q_{x},\rho_{x}\right\}}_{x=1}^{2}$, respectively, where $p_{1}\neq q_{1}$. Suppose that in the MD of a quantum ensemble $S_{x}$, the guessing probability is $p_{\mathrm{guess}}^{\left(x\right)}$ and the minimum value of guessing probability is $p_{\mathrm{guess}}^{★}$. Then, when $p_{\mathrm{guess}}^{\left(1\right)}=p_{\mathrm{guess}}^{\left(2\right)}$, if there exists an measurement that can simultaneously perform MD on two quantum ensembles $S_{1}$ and $S_{2}$, one can obtain $p_{\mathrm{guess}}^{\left(1\right)}=p_{\mathrm{guess}}^{★}$.*


Note that the optimal measurement performing simultaneous MD on two quantum ensemble $S_{1}$ and $S_{2}$ satisfies the equal probabilities of correct detection. It is the minimax strategy of the receiver. Here, the set of prior probability providing the minimum of guessing probability is a convex set. It can be shown in the following way. Suppose that the prior probabilities of q and p provide the minimum of guessing probability $p_{\mathrm{guess}}^{★}$. Then, by Lemma 3 there exists a measurement M that can perform MD on both the quantum states, satisfying $\sum_{x=1}^{2}q_{x}\mathrm{tr}(\rho_{x}M_{x})=p_{\mathrm{guess}}^{★}=\sum_{x=1}^{2}p_{x}\mathrm{tr}(\rho_{x}M_{x})$. Now, one can see that the relation of $\sum_{x=1}^{2}\left(\theta q_{x}+\left(1-\theta \right)p_{x}\right)\mathrm{tr}(\rho_{x}M_{x})=p_{\mathrm{guess}}^{★}$ holds for θ∈[0,1]. If one assumes that the minimax strategy (q,M) is not unique and there is another strategy p for a sender, then the minimax strategy of the sender forms a convex set, and one can find the prior probability where M is optimal in the ϵ-neighborhood of q for an arbitrary positive number of ϵ. Therefore, one can check the uniqueness of the prior probability of q providing the minimum of guessing probability, by deciding whether there exists a prior probability exhibiting optimal M in the ϵ neighborhood of q after finding the optimal POVM M for minimax strategy in the prior probability of q providing the minimum of guessing probability. Proposition 1 shows the necessary and sufficient condition for the non-uniqueness of a prior probability q of which M is optimal in the ϵ neighborhood (The proof of this proposition can be found in the Appendix A).

**Proposition** **1.**
*The prior probability providing the minimum of guessing probability is not unique if and only if ${\left\{M_{x}\right\}}_{x=1}^{2}$ satisfies the following conditions.*
*1*.
*$[\rho_{x},M_{1}]=0$∀x∈{1,2},*
*2*.
*For some x∈{1,2}, every $|v\rangle \in \mathrm{Supp}(M_{x})$ satisfies $\langle v|\rho_{1}|v\rangle :\langle v|\rho_{2}|v\rangle \neq 1-q:q$.*

*where [A,B]=AB−BA.*


Lemma 2 and Proposition 1 can be applied to check whether the strategy under a situation is unique. By applying Lemma 2, one can explain why the identical prior probability in the BB84 protocol is the best strategy of a sender. The quantum states used in the BB84 protocol are {|0〉,|1〉,|+〉,|−〉} [23]. Alice encodes $\left(a_{0},a_{1}\right)$ into quantum states and sends them to Bob. In general, $a_{0}$ is selected by Alice, but $a_{1}$ is randomly chosen. Suppose that Table 1 is used for encoding bit. Here, encoding means that a quantum state corresponding to $a_{0}a_{1}$ is prepared for communication.

When Alice chooses 0 as the value of $a_{0}$, the quantum state is determined by $a_{1}$. If the value of $a_{1}$ is 0, the quantum state becomes |0〉. However, when $a_{1}$ is 1, |1〉 is prepared for the quantum state. If the quantum state does not interact with the environment, Bob receives the quantum state prepared by Alice. Then, Bob performs the following measurements:(6)$$M_{0}=\left\{|0\rangle \langle 0|,|1\rangle \langle 1|\right\},\;M_{1}=\left\{|+\rangle \langle +|,|-\rangle \langle -|\right\}$$
When for the quantum states {|0〉,|1〉,|+〉,|−〉} the prior probability of the quantum states is identical, the optimal measurement becomes
(7)$$M=\{\frac{1}{2}|0\rangle \langle 0|,\frac{1}{2}|1\rangle \langle 1|,\frac{1}{2}|+\rangle \langle +|,\frac{1}{2}|-\rangle \langle -|\}.$$
The optimal measurement satisfies the following relation:(8)$$\mathrm{tr}(|0\rangle \langle 0|\frac{1}{2}|0\rangle \langle 0|)=\mathrm{tr}(|1\rangle \langle 1|\frac{1}{2}|1\rangle \langle 1|)=\mathrm{tr}(|+\rangle \langle +|\frac{1}{2}|+\rangle \langle +|)=\mathrm{tr}(|-\rangle \langle -|\frac{1}{2}|-\rangle \langle -|)=0.5$$
It implies that the identical prior probability provides the minimum of guessing probability. It is because if there exists an optimal measurement satisfying the above condition, the prior probability provides the minimum of guessing probability. It should be noted that Lemma 3 implies that the measurement of *M* is optimal for every prior probability providing the minimum of guessing probability. Meanwhile, if the prior probability is not identical, all the quantum states in the BB84 protocol does not have *M* as an optimal measurement. It can be shown in the following way. Let us assume that the probability of $a_{1}$ to become 0 or 1 is equal, and the probability of $a_{0}$ to become 0 or 1 is *q*. Then, the prior probability of each quantum state becomes q/2,q/2,(1−q)/2,(1−q)/2. We can show that if the measurement *M* is optimal at q≠0.5, the following inequalities should be satisfied [2,28]:(9)$$\begin{array}{c} \frac{q}{4}|0\rangle \langle 0|+\frac{q}{4}|1\rangle \langle 1|+\frac{1-q}{4}|+\rangle \langle +|+\frac{1-q}{4}|-\rangle \langle -|-\frac{q}{2}|0\rangle \langle 0|\geq 0 \end{array}$$(10)$$\begin{array}{c} \frac{q}{4}|0\rangle \langle 0|+\frac{q}{4}|1\rangle \langle 1|+\frac{1-q}{4}|+\rangle \langle +|+\frac{1-q}{4}|-\rangle \langle -|-\frac{1-q}{2}|+\rangle \langle +|\geq 0 \end{array}$$
However, when q≠0.5, one of these inequalities cannot be satisfied. Therefore, the prior probability providing the minimum of the guessing probability is only the case of q=1/2.

Using Proposition 1, we can investigate the quantum states of the unique prior probability, which provides the minimum of the guessing probability. Here we consider the MD of the following two quantum states:(11)$$\begin{array}{cc} \rho_{1} & =\frac{2}{3}|\phi^{-}\rangle \langle \phi^{-}|+\frac{I}{12}, \end{array}$$
(12)$$\begin{array}{cc} \rho_{2} & =\frac{1}{3}|\phi^{-}\rangle \langle \phi^{-}|+\frac{I}{6}. \end{array}$$

From Figure 2, we can check whether the prior probability providing the minimum of guessing probability is unique. We can see that the prior probabilities providing the minimum of guessing probability are $q_{1}=\frac{2}{5}$ and $q_{2}=\frac{3}{5}$. The optimal measurement for the quantum ensemble is $\left\{M_{1}=\frac{4}{5}|\phi^{-}\rangle \langle \phi^{-}|,M_{2}=I-\frac{4}{5}|\phi^{-}\rangle \langle \phi^{-}|\right\}$, since the measurement satisfies Lemma 1 as follows:(13)$$\begin{array}{cc} \left(q\rho_{1}-\left(1-q\right)\rho_{2}\right)M_{1} & =\left(\frac{2}{5}\rho_{1}-\frac{3}{5}\rho_{2}\right)M_{1}=\left(\frac{1}{15}|\phi^{-}\rangle \langle \phi^{-}|-\frac{1}{15}I\right)\frac{4}{5}|\phi^{-}\rangle \langle \phi^{-}|=0 \end{array}$$
(14)$$\begin{array}{cc} \left(\left(1-q\right)\rho_{2}-q\rho_{1}\right)M_{2} & =\left(\frac{3}{5}\rho_{2}-\frac{2}{5}\rho_{1}\right)M_{2}\\ & =\left(\frac{1}{15}I-\frac{1}{15}|\phi^{-}\rangle \langle \phi^{-}|\right)\left(I-\frac{4}{5}|\phi^{-}\rangle \langle \phi^{-}|\right)=\frac{1}{15}\left(I-|\phi^{-}\rangle \langle \phi^{-}|\right)\geq 0 \end{array}$$
In addition, ${\left\{M_{x}\right\}}_{x=1}^{2}$ satisfies the relation of $\mathrm{tr}(\rho_{1}M_{1})=\mathrm{tr}(\rho_{1}-\frac{4}{5}|\phi^{-}\rangle \langle \phi^{-}|)=0.6=\mathrm{tr}(\rho_{2}\left(I-\frac{4}{5}|\phi^{-}\rangle \langle \phi^{-}|\right))=\mathrm{tr}(\rho_{2}M_{2})$. From Lemma 2, the prior probability of $q_{1}=\frac{2}{5}$ and $q_{2}=\frac{3}{5}$ provides the minimum of guessing probability. Now, we verify the uniqueness of the prior probability which provides the minimum of the guessing probability for the given quantum states. The following relations show $\left[\rho_{1}+\rho_{2},M_{1}\right]=\left[\rho_{2},M_{1}\right]=0$, which is the first condition of Proposition 1:(15)$$\begin{array}{c} \left(\rho_{1}+\rho_{2}\right)M_{1}=\left(|\phi^{-}\rangle \langle \phi^{-}|+\frac{I}{4}\right)|\phi^{-}\rangle \langle \phi^{-}|=|\phi^{-}\rangle \langle \phi^{-}|\left(|\phi^{-}\rangle \langle \phi^{-}|+\frac{I}{4}\right)=M_{1}\left(\rho_{1}+\rho_{2}\right) \end{array}$$(16)$$\begin{array}{c} \rho_{2}M_{1}=\left(\frac{1}{3}|\phi^{-}\rangle \langle \phi^{-}|+\frac{2}{3}\frac{I}{4}\right)|\phi^{-}\rangle \langle \phi^{-}|=|\phi^{-}\rangle \langle \phi^{-}|\left(\frac{1}{3}|\phi^{-}\rangle \langle \phi^{-}|+\frac{2}{3}\frac{I}{4}\right)=M_{1}\rho_{2} \end{array}$$
However, $|\phi^{-}\rangle$ which is the support of $M_{1}$ and $M_{2}$, satisfies the following relation:(17)$$\langle \phi^{-}|\rho_{1}|\phi^{-}\rangle :\langle \phi^{-}|\rho_{2}|\phi^{-}\rangle =\frac{9}{12}:\frac{6}{12}=\frac{3}{5}:\frac{2}{5}=1-q:q$$
Because the second condition of Proposition 1 cannot be satisfied, the prior probability providing the minimum of the guessing probability is unique.

Now, to investigate the case of non-unique prior probability, which provides the minimum of the guessing probability, we consider the following quantum states:$$\rho_{1}=\left(\begin{array}{cc} 0.3 & 0\\ 0 & 0.7 \end{array}\right),\;\rho_{2}=\left(\begin{array}{cc} 0.7 & 0\\ 0 & 0.3 \end{array}\right)$$
From Figure 2, we can see non-uniqueness of the prior probability, which can provide the minimum of guessing probability.

For $\rho_{1}$ and $\rho_{2}$ with the prior probability of q=0.5, we can obtain the minimum of the guessing probability, which is 0.7. Then, the optimal measurements are $M_{1}=\left(\begin{array}{cc} 0 & 0\\ 0 & 1 \end{array}\right)\;and\;M_{2}=\left(\begin{array}{cc} 1 & 0\\ 0 & 0 \end{array}\right)$. Because of $\mathrm{tr}(\rho_{1}M_{1})=\mathrm{tr}(\rho_{2}M_{2})=0.7$, the minimum of the guessing probability becomes 0.7 at q=0.5. And because of $\left(\rho_{1}+\rho_{2}\right)M_{1}=M_{1}$, the support of $M_{1}$ is $|e_{2}\rangle ={\left(0,1\right)}^{T}$, which is unique. Then, one has
$$\langle e_{2}|q\rho_{1}-\left(1-q\right)\rho_{2}|e_{2}\rangle =\langle e_{2}|\frac{1}{2}\left(\begin{array}{cc} -0.4 & 0\\ 0 & 0.4 \end{array}\right)|e_{2}\rangle =0.2>0.$$
Further, because of $\left(\rho_{1}+\rho_{2}\right)M_{2}=M_{2}$, the support of $M_{2}$ is $|e_{1}\rangle ={\left(1,0\right)}^{T}$, which is unique. We have
$$\langle e_{1}|q\rho_{1}-\left(1-q\right)\rho_{2}|e_{1}\rangle =\langle e_{1}|\frac{1}{2}\left(\begin{array}{cc} -0.4 & 0\\ 0 & 0.4 \end{array}\right)|e_{1}\rangle =-0.2<0.$$
Then, the prior probability providing the minimum of the guessing probability is not unique.

From Proposition 1, the unique prior probability providing the minimum of the guessing probability has two cases. The interesting case of the two cases is one where the prior probability providing the minimum of the guessing probability is unique, with the condition that the second inequality of Proposition 1 does not hold. This is because, in this case, Bob’s optimal MD strategy is not unique. When the second condition of Proposition 1 is satisfied, an element $|v_{1}\rangle$ in the support of $M_{1}$ satisfies the relation of $\langle v_{1}|\rho_{1}|v_{1}\rangle :\langle v_{1}|\rho_{2}|v_{1}\rangle =1-q:q$. Then, from Lemma A1 in Appendix B, there exists ϵ>0 providing $M_{1}-\epsilon |v_{1}\rangle \langle v_{1}|\geq 0$. Now, we define $M_{1}^{\prime }$ and $M_{2}^{\prime }$ as $M_{1}-\epsilon |v_{1}\rangle \langle v_{1}|$ and $M_{2}+\epsilon |v_{1}\rangle \langle v_{1}|$, respectively. Then, $M_{1}^{\prime }$ and $M_{2}^{\prime }$ are positive semidefinite operators. Because of $M_{1}^{\prime }+M_{2}^{\prime }=\left(M_{1}-\epsilon |v_{1}\rangle \langle v_{1}|\right)+\left(M_{2}+\epsilon |v_{1}\rangle \langle v_{1}|\right)=I$, $M^{\prime }=\left(M_{1}^{\prime },M_{2}^{\prime }\right)$ is a POVM. We can verify whether $M^{\prime }$ is an optimal measurement at q. First, from the relation of $\langle v_{1}|\rho_{1}|v_{1}\rangle :\langle v_{1}|\rho_{2}|v_{1}\rangle =1-q:q$, we have $\langle v_{1}|\left(1-q\right)\rho_{2}-q\rho_{1}|v_{1}\rangle =0$. For $|v_{1}\rangle \in \mathrm{Supp}(M_{1})$, by Lemma A2, we can obtain $\left(\left(1-q\right)\rho_{2}-q\rho_{1}\right)|v_{1}\rangle \langle v_{1}|=0$. Then, we have the following relations that show that $M^{\prime }$ is an optimal measurement at q:$$\begin{array}{c} \left(\left(1-q\right)\rho_{2}-q\rho_{1}\right)M_{1}^{\prime }=\left(\left(1-q\right)\rho_{2}-q\rho -1\right)\left(M_{1}-\epsilon |v_{1}\rangle \langle v_{1}|\right)=\left(\left(1-q\right)\rho_{2}-q\rho_{1}\right)M_{1}\geq 0\\ \left(q\rho_{1}-\left(1-q\right)\rho_{2}\right)M_{2}^{\prime }=\left(q\rho_{1}-\left(1-q\right)\rho_{2}\right)\left(M_{2}+\epsilon |v_{1}\rangle \langle v_{1}|\right)=\left(q\rho_{1}-\left(1-q\right)\rho_{2}\right)M_{2}\geq 0 \end{array}$$
According to Lemma A3 in the Appendix B, the necessary and sufficient condition that two conditions of $[\rho_{1}+\rho_{2},M_{1}]=0$ and $[\rho_{2},M_{1}]=0$ are satisfied is that the optimal measurement M of a receiver can be simultaneously diagonalized with two quantum states $\rho_{1}$ and $\rho_{2}$. Therefore, if the optimal measurement M of a receiver is simultaneously diagonalized with two quantum states $\rho_{1}$ and $\rho_{2}$, the uniqueness of the sender’s minimax strategy cannot be compatible with the uniqueness of the receiver’s MD strategy. The following Corollary summarizes this result.

**Corollary** **1.**
*If the optimal measurement M can be simultaneously diagonalized with two quantum states $\rho_{1}$ and $\rho_{2}$, the uniqueness of minimax strategy of a sender and the uniqueness of MD of a receiver cannot be compatible.*


The above result can be applied to cases of building quantum random number generator(QRNG). Suppose that only one side’s strategy is unique. Therefore, either the minimax strategy of a sender or the minimum error strategy is unique. The randomness in QRNG is defined as the min-entropy to the classical bit in the quantum-classical state and depends on the prior probability [25,29]. If the prior probability providing minimum guessing probability is not unique, we can build QRNG that is not sensitive to the prior probability. When QRNG is built such that the receiver’s strategy is unique, even a slight error in the measurement leads to the loss of the optimality of the receiver’s strategy. The quantum states with a unique receiver’s strategy in QRNG can be found by using Corollary 1.

## 4. Conclusions

We studied the two person zero sum game where the payoff is defined by the correct probability of the two quantum states. Because it is known that the optimal strategy of the game is a minimax strategy, and it is important to verify its uniqueness of the minimax strategy, we focused on the uniqueness condition of the minimax strategy of a sender and the minimax strategy of a receiver. In this study, we obtained the necessary and sufficient condition for the uniqueness of the sender’s strategy. Using this condition, we investigated the quantum states providing the minimum guessing probability when a sender’s minimax strategy is unique. Further, we found the condition where both the sender’s minimax strategy and the receiver’s optimal minimum error strategy cannot be unique.

Our result helps to understand the fundamental aspect of minimax strategy. We studied the minimax strategy in the quantum state discrimination of two quantum states. The uniqueness of the minimax strategy in the quantum state discrimination of more than two quantum states is not known yet. In our future work, we hope to investigate this problem.

## Figures and Tables

**Figure 1 entropy-21-00671-f001:**
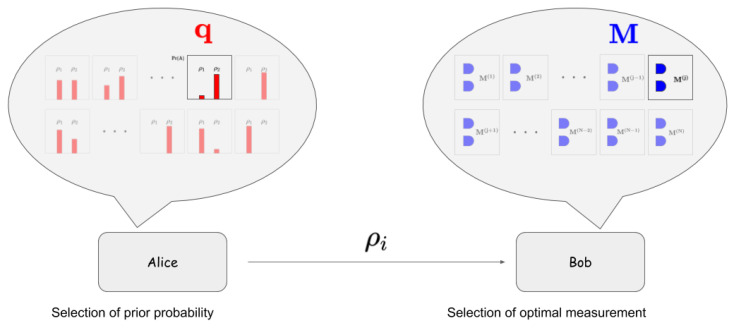
The strategy of the sender(Alice) and the receiver(Bob) in two-person zero-sum quantum game. The strategy of Alice is to choose the optimal prior probability q, which is the probability of quantum states prepared in the quantum system, to minimize the payoff. The strategy of Bob is to choose the optimal measurement to maximize payoff.

**Figure 2 entropy-21-00671-f002:**
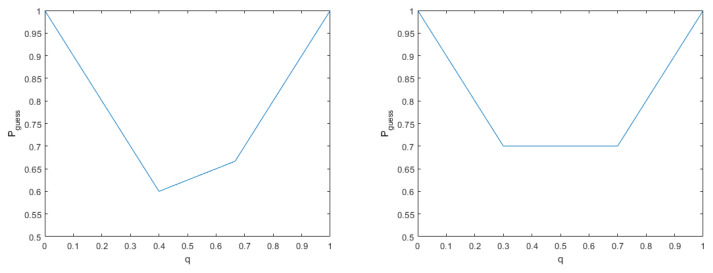
(**Left:**) Example of unique prior probability providing the minimum of guessing probability. The guessing probability of two quantum states $\rho_{1}=\frac{2}{3}|\phi^{-}\rangle \langle \phi^{-}|+\frac{I}{12}$ and $\rho_{2}=\frac{1}{3}|\phi^{-}\rangle \langle \phi^{-}|+\frac{I}{6}$ is shown in terms of prior probability (q,1−q). (**Right:**) Example of non-unique prior probability providing the minimum of guessing probability. The guessing probability of two quantum states $\rho_{1}=\mathrm{diag}\left[0.3,0.7\right]$ and $\rho_{2}=\mathrm{diag}\left[0.7,0.3\right]$ is shown in terms of prior probability (q,1−q).

**Table 1 entropy-21-00671-t001:** Encoding table for Alice.

$a_{0}a_{1}$	Quantum States
00	|0〉
01	|1〉
10	|−〉
11	|+〉

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
