# Peer review of "Uniqueness of Minimax Strategy in View of Minimum Error Discrimination of Two Quantum States"

_entropy, 2019, doi:10.3390/e21070671_

Round 1

Reviewer 1 Report

The present manuscript manuscript provides a detailed analysis of discrimination strategies for two, in general non-orthogonal, quantum states. The authors discuss such problem in the minimum discrimination scenario, and provide conditions for an optimal strategy to be unique.

This paper is interesting and provides novel insights on discrimination strategies between quantum states. Before I can recommend this paper for publication, my main comment regards the introduction of the paper. More specifically, I would suggest the authors to significantly revise the introduction to provide a more clear picture of the previous results in the literature, so as to better clarify the novelty of this manuscript. For instance, the authors start by citing previous papers in the literature in a short first paragraph. Then, after a brief discussion on minimum error discrimation strategies, the authors directly rush on introducing their results. I would then suggest the authors to:

(i) extend the first paragraph, by providing a more detailed discussion on the previous strategies reported in the literature.

(ii) extend the second paragraph on MD strategies, in particular by adding here a definition of such class of approaches (which the reader may not be familiar with).

These changes would be of great help to any reader approaching the detailed results reported in the paper.

Author Response

Reviewer #1: The present manuscript provides a detailed analysis of discrimination strategies for two, in general non-orthogonal, quantum states. The authors discuss such problem in the minimum discrimination scenario, and provide conditions for an optimal strategy to be unique.

This paper is interesting and provides novel insights on discrimination strategies between quantum states. Before I can recommend this paper for publication, my main comment regards the introduction of the paper. More specifically, I would suggest the authors to significantly revise the introduction to provide a more clear picture of the previous results in the literature, so as to better clarify the novelty of this manuscript. For instance, the authors start by citing previous papers in the literature in a short first paragraph. Then, after a brief discussion on minimum error discrimation strategies, the authors directly rush on introducing their results. I would then suggest the authors to:

(i) extend the first paragraph, by providing a more detailed discussion on the previous strategies reported in the literature.

(ii) extend the second paragraph on MD strategies, in particular by adding here a definition of such class of approaches (which the reader may not be familiar with).

(1) Extend the first paragraph, by providing a more detailed discussion on the previous strategies reported in the literature.

Response: We thank you for the comment. According to your comments, we extended the first paragraph by providing a more detailed discussion on the known strategies. (The revised parts are underlined).

(before revision)

Quantum information processing can be achieved by discriminating quantum states, where classical information is encoded. Quantum states can be discriminated using many strategies[1-4], including minimum error discrimination(MD)[4-7], unambiguous discrimination[8-12], maximum confidence discrimination[13], and discrimination with fixed rate of inconclusive results[14-18]. Among these strategies, the MD strategy can conclusively discriminates quantum states with prior probability.

(after revision)

Quantum information processing can be achieved by discriminating quantum states, where classical information is encoded.Quantum states which are orthogonal to each other can be perfectly distinguishable. However, non-orthogonal quantum states cannot be perfectly discriminated. Therefore, one needs to have a discrimination strategy for non-orthogonal quantum states, and there are various strategies[1-4] such as minimum error discrimination(MD)[4-7], unambiguous discrimination[8-12], maximum confidence discrimination[13], and discrimination of fixed rate inconclusive result[14-18]. Unambiguous discrimination is a strategy where there is no error in the  conclusive result by allowing an inconclusive result. Maximum confidence is a strategy where one maximizes the confidence of a conclusive result.  Discrimination of fixed rate inconclusive result is a strategy where one may fix the rate of an inconclusive result. Among these strategies, the MD strategy can conclusively discriminate quantum states with a prior probability.

 (2) Extend the second paragraph on MD strategies, in particular by adding here a definition of such class of approaches (which the reader may not be familiar with).

Response: We appreciate your comment. According to your comment, we added the definition of minimum error discrimination and explained how our problem is related to the strategy of minimum error discrimination.(The revised parts are underlined.)

(before revision)

The MD strategy is employed for quantum states with a given prior probability and the quantum states are optimally measured. In fact, one can investigate the behavior of MD in terms of prior probability when quantum states are given. An optimal MD strategy provides guessing probability. One should note that the guessing probability is defined as the maximum probability to correctly discriminate among given quantum states through optimal measurement. Because the guessing probability is obtained based on prior probability, a change in prior probability will result in different guessing probabilities.

(after revision)

The MD strategy is employed for quantum states with a given prior probability, and the quantum states are optimally measured. MD strategy is that one maximizes the probability that the result of measurement of a receiver correctly points out the quantum state that a sender transmitted when an only conclusive result is permitted. The maximum probability is called guessing probability. One can investigate the behavior of MD in terms of a prior probability when quantum states are given.

Because the guessing probability is obtained based on prior probability, a change in prior probability results in different guessing probabilities, which implies that prior probabilities can be considered as a strategy of a sender. Even though one has discussed the uniqueness of measurement strategy in discrimination of two quantum states, the strategy of preparation such as a prior probability, which can be a strategy of a sender, has not been discussed in terms of identical guessing probability and optimal measurement strategy.

(3) Revise the introduction to provide a more clear picture of the previous results in the literature, so as to better clarify the novelty of this manuscript.

Response: We thank you for the comment. According to your comment, we explained previous results and clarified the novelty of our work.(The revised parts are underlined.)

(addition)

Quantum minimax approach is obtained by applying the minimax approach of a statistical decision to quantum state discrimination. Von Neumann, the inventor of game theory, showed that there exists a solution to the minimax problem when sender and receiver can choose a finite number of strategy in a two-person zero-sum game. Wald proved that the necessary and sufficient condition to the existence of a solution to the minimax problem is that the set of strategy for sender and receiver is countable[19]. Hirota and Ikehara discussed quantum minimax theorem, using the fact that the set of measurement strategy satisfies compactness[20]. They suggested the necessary and sufficient condition for minimax strategy in quantum state discrimination.

Further, by mean value theorem D’Ariano showed that there exists a quantum minimax strategy for two quantum state discrimination and provided a sufficient condition for the strategy[21]. However, in spite of these studies, the necessary and sufficient condition for uniqueness of minimax strategy in two quantum state discrimination is not known yet. Even more, the uniqueness of minimax strategy in two quantum state discrimination is not understood in terms of sender’s strategy, which is a selection of prior probability.

Reviewer 2 Report

This is a nice paper adding what seems to be new and useful results on uniqueness of an optimal strategy for quantum state discrimination. The material in the paper is very well organised and nicely presented (clear theory, nice examples, ...). I give it mainly the score "average" rather than "high" for novelty, significance etc, because I think that the results are not very surprising and they seem fairly easy to obtain. This is a useful completion of a large body of existing work. I only have one problem with the paper, and that is the English. The authors again and again make an unfortunate choice of "article" (a/the/"space") in such a way that I have to read and re-read every sentence multiple times in order to figure out what the authors are saying. There are also many more, on the whole quite obvious, grammatical errors. The authors badly need to have the paper carefully read by a native English speaker. The reader will have to talk to the authors in person, in order to find out what they want to say! 

I am sure that modern software tools could also help a great deal in fixing grammatical errors. The errors with the definite or indefinite article are not a question of grammar (syntax) but of content. The meaning changes according to which article (if any) you use. I realise that these differences of meaning are hard for people to appreciate whose native language is of such different nature as, in this case, Korean. But too bad ... that's the way it goes. The paper is written in "English" and the authors need to gain English writing skills if they want to communicate effectively. In my case: I can read the math formulas and therefore find out what is the real content of the paper. But if the authors want readers to be attracted to the paper they must make sure that especially the abstract, introduction, and conclusion, are written in *perfect* English. People are only going to put energy into reading the core parts of the paper, if they have already *easily* learnt from the outer parts that this is going to be worth their while.

Author Response

Reviewer #2: This is a nice paper adding what seems to be new and useful results on uniqueness of an optimal strategy for quantum state discrimination. The material in the paper is very well organised and nicely presented (clear theory, nice examples, ...). I give it mainly the score "average" rather than "high" for novelty, significance etc, because I think that the results are not very surprising and they seem fairly easy to obtain. This is a useful completion of a large body of existing work. I only have one problem with the paper, and that is the English. The authors again and again make an unfortunate choice of "article" (a/the/"space") in such a way that I have to read and re-read every sentence multiple times in order to figure out what the authors are saying. There are also many more, on the whole quite obvious, grammatical errors. The authors badly need to have the paper carefully read by a native English speaker. The reader will have to talk to the authors in person, in order to find out what they want to say!

Response: We appreciate your comments. According to your comments, we corrected grammatical errors and corrected the wrong choice of "article". Further, our paper was revised by a native English speaker.

Entropy EISSN 1099-4300 Published by MDPI AG, Basel, Switzerland RSS E-Mail Table of Contents Alert
Back to Top